# Effects of *Gryllus bimaculatus* and *Oxya chinensis sinuosa* extracts on brain damage via blood-brain barrier control and apoptosis in mice with pentylenetetrazol-induced epilepsy

**Ngoc Buu Tran**⊙**, Sook-Jeong Lee**⊙*

Department of Bioactive Materials Sciences and Research Center of Bioactive Materials, Jeonbuk National University, Jeonju, Jeollabuk-do, Korea

* sj@jbnu.ac.kr

**Data Availability Statement:** All relevant data are within the paper and its Supporting Information files.

## Abstract

The demand for environmentally friendly foods with high nutritional value and low carbon emissions is increasing with the aging of the global population and the crisis of food resources. Edible insects are becoming increasingly well-known as such foods. This study evaluated the effects and mechanisms of *Gryllus bimaculatus* (Cricket) (*Gb*) and *Oxya chinensis sinuosa* (Grasshopper) (*Ocs*) extracts on epilepsy. A pentylenetetrazol (PTZ)-induced seizure mouse model was used for the study, and *Gb* and *Ocs* extracts were administered for 29 days on alternate days at concentrations of 8 g/kg and 16 g/kg. The integrity of the blood-brain barrier (BBB) and brain edema was measured using the perfusion of Evans blue dye and brain water content. *Gb* and *Ocs* extracts prevented BBB permeabilization and cerebral edema through increasing the expression of tight junction-associated proteins in the endothelial cells and reducing water content in PTZ-treated mice. Additionally, *Gb* and *Ocs* extracts protected neurons from oxidative stress and apoptosis in different brain areas. These protective effects were demonstrated through the restoration of the expression of neuronal nuclear protein and postsynaptic density protein-95, thus increasing the levels of glutathione and superoxide dismutase, decreasing lipid peroxidation, and recovering apoptosis-associated proteins, such as Bax, cleaved PARP, and cleaved caspase-3, in epileptic mice. In addition, *Gb* and *Ocs* extracts rescued PTZ-induced hyperexcitable neurons to control mice level, as supported by the restored expression of gamma-aminobutyric acid (GABA) transporter 1, the metabotropic glutamate receptors–GRM2/3, and BDNF. This study suggested that *Gb* and *Ocs* extracts are novel medicinal candidates that can help ameliorate epilepsy by improving BBB health and preventing oxidative stress-mediated apoptosis.

## Introduction

Based on the climate crisis caused by the environmental pollution, a global interest in high-nutritional foods with a low carbon footprint, such as meat substitutes and edible insects, is

**Funding:** This study was financially supported by Ministry of Science and ICT (MSIT, Republic of Korea) (https://www.msit.go.kr/) in the form of a National Research Foundation (NRF) grant (NRF-2021R1A2C1005980) awarded to S-JL. This study was also financially supported by Ministry of Science and ICT (MSIT, Republic of Korea) via Commercialization Promotion Agency for R&D Outcomes (COMPA, Republic of Korea) (https://www.compa.re.kr/) in the form of a "Laboratory-specialized Start-up Leading University Project" grant (RS-2023-00254730) awarded to S-JL. No additional external funding was received for this study. The funders had no role in study design, data collection and analysis, decision to publish, or preparation of the manuscript.

**Competing interests:** The authors have declared that no competing interests exist.

increasing to help cope with the food crisis. Edible insects are rich in various nutrients, including minerals, vitamins, fibers, and unsaturated fatty acids [1], comprise abundant antioxidants, and exhibit neuroprotective properties [2,3]. Recently, numerous studies have reported the medicinal effects of edible insects on different diseases, including metabolic diseases such as diabetes, cancer, and neuropsychiatric diseases [2–4].

Epilepsy is a common neurological disorder, with the most prominent feature being recurrent unprovoked seizures [5,6], which are primarily accompanied by blood-brain barrier (BBB) dysfunction, brain injury, tumor, hypoxia, and dysregulation of ion channels and neurotransmitter receptors, contributing to excitatory/inhibitory imbalances [7,8]. However, mechanisms underlying the development of epileptic seizures remain unclear.

Oxidative stress is responsible for neuronal loss and death, which are directly related to epilepsy in *in vitro* animal models [9,10], and disturbs the antioxidant systems, including catalase (CAT), glutathione (GSH), superoxide dismutase (SOD), and lipid peroxidation (malondialdehyde [MDA]) systems [11,12]. Excess reactive oxygen species (ROS) activate pro-apoptotic signaling cascades, resulting in apoptotic cell death [13]. Apoptosis is a process that involves cell death with common functions such as cell replacement, removal of damaged cells, and tissue regeneration [14]. The Bcl-2 family of proteins, which is important for regulating the apoptotic process, has been classified into pro-apoptotic proteins, including Bax and BID, and anti-apoptotic proteins, including Bcl-2 and Bcl-XL [15]. Additionally, caspase-3 and 9, which are cysteine proteases, are involved in this process [15].

The BBB is a highly selective, semi-permeable biological barrier characterized by the active control of circulating blood solutes that pass non-selectively into the intracerebral environment, with significant effects on nerve cells [16,17]. The BBB is extremely vulnerable to oxidative stress and can be damaged by the ROS. Previous studies have suggested that BBB injury plays an important role in the molecular mechanisms underlying various neurological disorders and neurodegenerative diseases. Impaired BBB functions facilitate the formation of toxic substances that damage nerve cells, i.e., the disruption of BBB integrity has been reported in patients with Alzheimer's disease (AD), vascular cognitive impairment [18], Parkinson's disease (PD), toxin-induced animal PD models [19], and epilepsy models induced by pentylenetetrazol (PTZ) [20,21].

To the best of our knowledge, no studies have yet reported the effects of insect extracts, an environmentally friendly biomaterial that can cope with the climate crisis, on BBB damage and synaptic plasticity. Thus, our study aims to determine the effects of the extracts of the edible insects, *Gryllus bimaculatus* (*Gb*) and *Oxya chinensis sinuosa* (*Ocs*), in a mouse model of PTZ-induced epilepsy. In particular, we focused on the effects of *Gb* and *Ocs* extracts on BBB permeabilization and oxidative stress-triggered neuronal apoptosis.

## Materials and methods

### Chemicals and antibodies

PTZ (#P6500-25G), Evans blue (#E2129), and toluidine blue (#89640) were purchased from Sigma Aldrich (St. Louis, MO, USA). *Gb* and *Ocs* were obtained from Purnae Company (Sejong, Korea). The anti-AChE antibody (#MBS9605181) was purchased from MyBioSource, Inc. (San Diego, CA, USA), ZO-1 (#sc-33725) from Santa Cruz Biotechnology Inc. (Dallas, TX, USA), and anti-claudin-5 antibody (#MA5-32614) from Thermo Fisher Scientific (Waltham, MA, USA). Anti-ChAT (#ab178850), the neuronal marker (anti-neuronal nuclear protein anti-Neu-N) (#ab104224), the synaptic marker anti-PSD-95 (#ab192757), anti-matrix metalloproteinase (MMP)-2 (#ab86607), and lipid peroxidation (MDA) assay kits (#ab118970) were purchased from Abcam (Cambridge, UK). Occludin (#E6B4R), Bax (#2772), Bcl-XL

(#2764), cleaved (c)-PARP (#5625T), and c-caspase-3 (#9664T) antibodies were purchased from Cell Signaling Technology (Danvers, MA, USA). The anti-brain-derived neurotrophic factor (BDNF) (#GTX132621) and the anti-gamma-aminobutyric acid (GABA) transporter 1 (GAT1) (#GTX133150) antibodies were obtained from GeneTex Inc. (Alton Pkwy, Irvine, CA, USA). The anti-metabotropic GluR2/3 (GRM2/GRM3) (#CSB-PA009022) antibody was purchased from Cusabio Technology (Huston, TX, USA). $\alpha$-Tubulin (#BS1699) antibody was obtained from Bioworld Technology (Nanjing, China). The EnzyChrom$^{TM}$ GSH/GSSG Assay Kit (#EGTT-100) was purchased from BioAssay Systems (Hayward, CA, USA). The SOD Activity Assay Kit (#K335-100) was purchased from BioVision, Inc. (Waltham, MA, USA).

## Construction of the PTZ-induced epilepsy mouse model and epileptogenesis assessment

Eight-week-old male ICR mice (weight, 20–25 g) were purchased from Samtaco (Suwon, Korea) and were maintained at 24±2˚C and 50±5% humidity under a 12-h light/dark cycle. After the mice were acclimatized for a week, all animal experiments were conducted after obtaining the approval of the Animal Experimentation Ethics Committee of Jeonbuk National University [JBNU 2021–071].

To determine the optimal subconvulsive concentration of PTZ for epilepsy induction, we divided the mice into five groups and administered different concentrations of PTZ: (1) control group (n = 8), (2) PTZ 20 mg/kg (n = 8), (3) PTZ 40 mg/kg (n = 8), (4) PTZ 60 mg/kg (n = 8), and (5) PTZ 80 mg/kg (n = 8). The dose with the maximum efficacy in epilepsy induction was chosen for further experiments. The mice were divided into seven groups of eight animals each (n = 56). To induce epilepsy, all mice in each group were administered with intraperitoneal (i.p.) PTZ as follows: (1) the control group received 0.9% saline and 2% DMSO (n = 10); (2) the seizure group received PTZ (40 mg/kg) (n = 10); (3) the positive control group received PTZ and valproic acid (VPA) (100 mg/kg) (n = 10), an anti-epilepsy drug; (4) PTZ 40 mg/kg and 8 g/kg *Gb* (n = 10); (5) PTZ 40 mg/kg and 16 g/kg *Gb* (n = 10); (6) PTZ 40 mg/kg and 8 g/kg *Ocs* (n = 10); and (7) PTZ 40 mg/kg and 16 g/kg *Ocs* (n = 10).

As shown in Fig 1, regarding the combination drug treatment, VPA or insect extract was administered 30 min before PTZ was administered. To induce kindling, we injected mice with a subconvulsive dose of PTZ (40 mg/kg, i.p.) or VPA (100 mg/kg, i.p.) on alternate days for 29 days, with excluding mice in the control group (Fig 1). The experimental design was based on

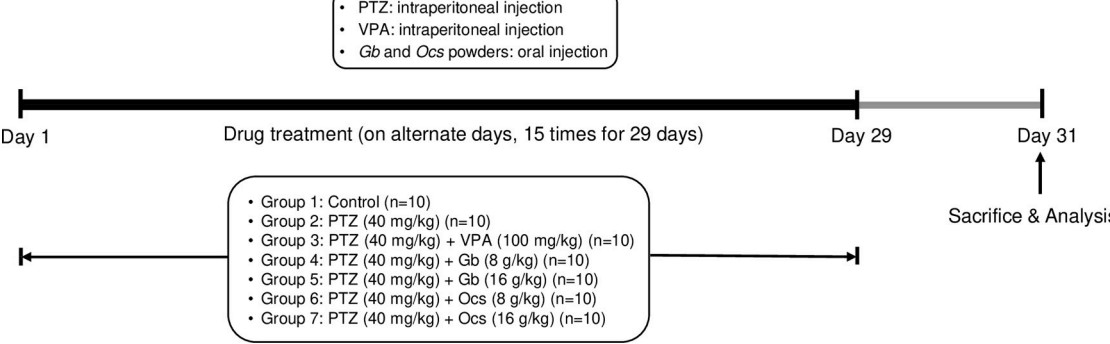

**Fig 1. Schematic diagram of the experimental design of the pentylenetetrazol (PTZ)-induced epilepsy model, drug treatment, and biochemical analysis in mice.** PTZ was administered intraperitoneally (i.p.) to the mice on alternating days (15 injections over 29 days). VPA (i.p.), *Gryllus bimaculatus* (*Gb*), and *Oxya chinensis sinuosa* (*Ocs*) extracts were administered orally 30 min before initiating the PTZ treatment.

**Table 1. Scoring of pentylenetetrazol-induced seizure in mice.**

| Score | Behavioral Characteristics |
|---|---|
| 1 | Immobilization or no convulsive activity |
| 2 | Head nodding or twitching of ear facial muscles |
| 3 | Myoclonic twitches |
| 4 | Rearing, forelimb clonus |
| 5 | Jumping, generalized-clonic seizures, or falling |
| 6 | Tonic hind limb extensions or tonic-clonic convulsions |

an established epileptic model with minor changes [22]. Dried *Gb* and *Ocs* were extracted using 10 volumes of distilled water at 60˚C for 12 h. The extracts were filtered, concentrated under reduced pressure using a vacuum evaporator (IKA™ RV 3V Rotary Evaporator, Thermo Fisher Scientific), and freeze-dried. The prepared samples were stored at 4˚C until future use. Specifically, the mice in groups 4, 5, 6, and 7 received 0.1 mL *Gb* and *Ocs* extracts (orally, once a day) for 4 weeks from day 1 (D1) to D29, and the control group was received water of the same amount. The drugs were freshly prepared immediately before the administration. The experimental setup is illustrated in Fig 1.

Epileptogenesis was evaluated through measuring the seizure score and was graded from 1 to 6 according to previously established protocols [23]. Epileptogenesis was induced using a subconvulsive dose of PTZ (40 mg/kg, i.p.) on alternate days (14 injections over 29 days) [24]. Immediately after each injection of PTZ, mice in all groups were examined for seizures for 30 min for establishing a seizure score by an observer who was blinded to the treatment status (Table 1). Important brain regions such as the prefrontal cortex (PFC) and hippocampus (HC) were evaluated for the effects of insect powders, *Gb*, and *Ocs* in a model of epilepsy induced by PTZ.

## Evans blue permeability and brain water content measurement

Evans blue content in the brain indicates the extent of the BBB damage [25]. Briefly, the mice were injected with 2% Evans blue dye in 0.9% saline (2 mL/kg, i.p.), which was maintained for 2 h. They were then anesthetized and injected with saline through the heart to flush any remaining dye from the blood vessels before obtaining brain samples for conducting the qualitative evaluation of Evans blue-stained sections. In addition, these brain samples were weighed and used to determine the Evans blue content via spectrophotometry. Evans blue dye was extracted by homogenizing each brain tissue sample in phosphate-buffered saline at a pH of 7.4. Proteins were precipitated by adding 6 mL of 60% trichloroacetic acid. The absorbance of the supernatant was measured at 610 nm via an Epoch™ Microplate Spectrophotometer (Bio-Tek Instruments, Winooski, VT, USA). Finally, the Evans blue content (mg of Evans blue/g of tissue) in the brain was determined based on a standardized curve.

Moreover, brain water content was evaluated. The mice were then anesthetized and euthanized. The brains were collected, weighed (wet weight), and placed in a drying oven (MMM Medcenter™ Venticell™ Drying Ovens, Germany) at 60˚C for 72 h. Subsequently, the brains were weighed again (dry weight). The brain water content was determined as the difference between the wet tissue weight and dry weight [26,27]. It was calculated as follows:

$$Brain\ water\ content = \frac{(\text{Wet weight} - \text{Dry weight})}{\text{Wet weight}} \times \frac{100}{\text{animal body weight}}$$

### Western blot analysis

Brain tissue was collected, and the prefrontal cortex (PFC) and hippocampus (HC) were separately isolated from the brain and homogenized in radioimmunoprecipitation assay buffer (150 mM sodium chloride, 1% Triton X-100, 0.5% sodium deoxycholate, 0.1% sodium dodecyl sulfate, and 50 mM Tris; pH 8.0). The protein concentrations were measured via the bicinchoninic acid kit (Pierce™ BCA Protein Assay Kit, Thermo Fisher Scientific). Equal amounts of proteins were separated via sodium dodecyl sulfate-polyacrylamide gel electrophoresis and were transferred to polyvinylidene difluoride membranes. Immunoreactive proteins were visualized using an enhanced chemiluminescence kit (Thermo Scientific™ West Femto maximum sensitivity substrate, #34095) and iBright CL1000 imaging system (Thermo Fisher Scientific). Protein expression was quantified using a densitometric analysis of the intensity of each protein band via the ImageJ software and was normalized to the corresponding α-tubulin bands or each total protein band.

### Histopathological analysis

Neurological injury and morphological changes in the PFC and HC brain regions were assessed using histological analyses. The HC regions were observed in the CA1, CA3, and dentate gyrus (DG). Briefly, 20-$\mu$m-thick coronal sections of the brain were obtained using a cryostat (CM 1510S; Leica Microsystems, Germany) and were mounted on slides. The sections were deparaffinized using xylene and were rehydrated using ethanol at 100%, 90%, 70%, and 50%. They were then stained with toluidine blue solution for 10 min, rapidly rinsed in distilled water, and dehydrated in 95% ethanol for 20 min. Digitized images of Nissl staining were obtained via an optical microscope (Nikon Eclipse Ts2) equipped with a camera (HK6E3 E3CMOS; Nikon Inc., Tokyo, Japan), using the same settings for all samples.

### Oxidative stress marker activities assessment

The levels of GSH, SOD, and MDA activity in the whole-brain extracts at D31 were analyzed using a microplate reader (Epoch™ Microplate Spectrophotometer, Bio-Tek Instruments) at 412 nm, 450 nm, and 532 nm, respectively. The manufacturer's instructions for commercially available kits using this mechanism were followed as previously reported [11,12].

### Data analysis

All data were presented as mean ± standard deviation. For multiple comparisons among groups, we used a one-way or two-way analysis of variance, followed by Fisher's least significant difference post hoc tests. Paired $t$-tests were used to analyze the differences between the two groups. Statistical significance was set at $P$-values of $< 0.05$.

## Results

### Effects of PTZ, *Gb*, and *Ocs* extracts on seizure scoring in mice

We conducted acute experiments using various concentrations of PTZ (0, 20, 40, 60, and 80 mg/kg) in mice and analyzed mortality at each dose (S1 Table) to select the optimal dose of PTZ for subconvulsive seizures. After the i.p. injection of PTZ, behavioral seizure scoring and video recordings were performed. PTZ injection resulted in epileptogenesis in a dose-dependent manner, with a mean seizure score of X (range: 2–6) (Fig 2A). Based on this result, we selected 40 mg/kg dose for subsequent chronic seizure experiments in mice based on a non-fatal mild seizure score of 3.5. Repeated injections of a subconvulsive dose of PTZ (40 mg/kg, i.p., once every other day 15 times for 29 days) (Fig 1) resulted in progressive development of

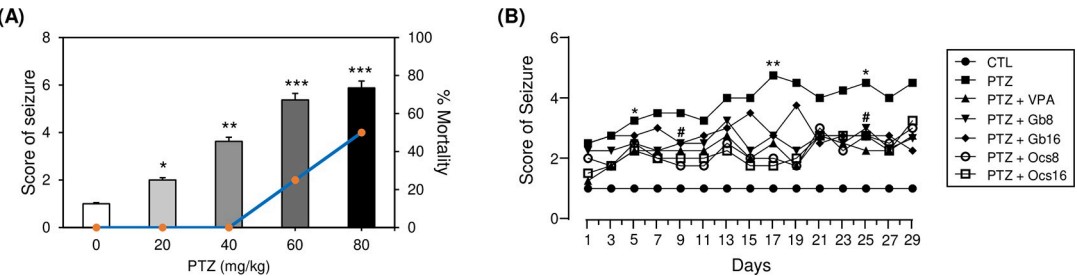

**Fig 2. Determination of subconvulsant PTZ kindling dose for seizure induction and effects of *Gb* and *Ocs* extracts on PTZ kindling-induced seizure severity in mice.** (A) Statistical results showing a PTZ dose-dependent increase in the seizure score. (B) A statistical analysis of seizure severity in the PTZ-induced epilepsy mouse model after administrating *Gb* and *Ocs* extracts. N = 4. Mean ± standard deviation (SD). *$P \leq 0.05$, **$P \leq 0.01$, and ***$P \leq 0.001$, vs. control group; #$P \leq 0.05$ vs. PTZ group.

seizures (Fig 2A and 2B). Therefore, we evaluated the effects of *Gb* and *Ocs* extracts (both at 8 and 16 mg/kg) after combining with 40 mg/kg PTZ on mortality (S2 Table). Similarly, to determine the optimal concentration of *Gb* and *Ocs* extracts to combine with PTZ, we decided four different doses of extracts (8, 16, and 20 g/kg) based on the concentrations known for use in animal models. After screening for seizure onset time, duration of seizure, and mortality, the 8- and 16-g/kg insect extracts were selected for further studies (S3 Table).

Monitoring the behavioral seizure stages revealed that the vehicle-treated mice exhibited no epileptogenesis (Fig 2B). In addition, the 40-mg/kg-PTZ-alone group showed the highest seizure score with the highest symptom severity after eight repeated PTZ injections (on D17 of the experiment) (Fig 2B). However, the seizure scores in the *Gb* (8 and 16 g/kg) and *Ocs* (8 and 16 g/kg) extract-treated mice were significantly lower than those in the PTZ-treated group (Fig 2B) and were dependent on the dose of *Gb* and *Ocs* extracts. The mice co-treated with VPA (100 mg/kg) and PTZ (40 mg/kg) were used as positive controls for seizure induction since VPA is an anti-seizure medication. The seizure score of this group was comparable to that of the group co-treated with PTZ, *Gb*, and *Ocs* (Fig 2B).

## Effects of *Gb* and *Ocs* extracts on BBB dynamics and brain edema in the PTZ-induced epilepsy mouse model

Evans blue cannot cross the BBB; thus, the accumulation of a significant amount of Evans blue indicates a severely damaged BBB, leading to many brain disorders. We reported that a 40-mg/kg PTZ injection noticeably increased the level of Evans blue penetration across the BBB compared with that in the control brains (Fig 3A). A quantitative analysis revealed that the intensity of Evans blue was the highest in the HC group, indicating that it was the most severely damaged region (Fig 3C), followed by the PFC, which was less vulnerable to PTZ (Fig 3B). Co-treatment with *Gb* and *Ocs* extracts significantly decreased the Evans blue level in a dose-dependent manner compared with that in the PTZ-alone group (Fig 3A–3C). In particular, compared with administrating PTZ alone, administrating a high concentration of *Ocs* (16 g/kg) resulted in the least BBB disruption (Fig 3A–3C).

Consistently, the markers of endothelial cell tight junctions, such as claudin-5, occludin, ZO-1, and MMP-2, were examined in the tissue lysates obtained from the PFC and HC. In both the PFC and HC of mice treated with PTZ alone, the expression levels of occludin, claudin-5, and ZO-1 were significantly reduced; however, the MMP-2 expression level was significantly higher than in the control group (Fig 3E and 3F). Furthermore, co-treatment with PTZ and either *Gb* or *Ocs* extract (8 and 16 g/kg) almost completely reversed the PTZ-mediated

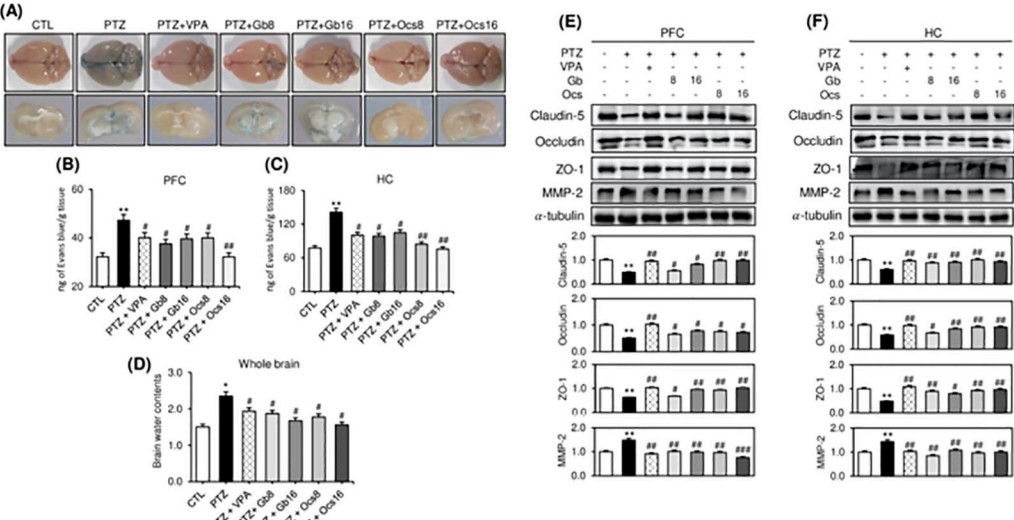

**Fig 3. Effects of *Gb* and *Ocs* extracts on blood-brain barrier integrity and brain edema in a PTZ-induced epilepsy mouse model.** (A) Representative photographs of Evans blue extravasation in the whole brain and individual hemispheres of various groups at day 31 (D31). (B-C) A quantitative analysis of Evans blue leakage from the PFC (B) and HC (C). (D) A quantitative analysis of brain edema based on brain water content measurements. (E-F) Immunoblots for ZO-1, claudin-5, occludin, and MMP-2 in the PFC I and HC (F) tissue lysates at D31 using different combination treatments. Bars denote a densitometric analysis of the immunoblots. Mean ± SD. $*P \leq 0.05$ and $**P \leq 0.01$ vs. control group; $^{\#}P \leq 0.05$, $^{\#\#}P \leq 0.01$, and $^{\#\#\#}P \leq 0.001$ vs. PTZ group.

alterations in the expression of claudin-5, occludin, ZO-1, and MMP-2 in a dose-dependent manner (Fig 3E and 3F).

In addition to BBB disruption, excess brain fluid is reported in patients with epilepsy. Herein, we examined the effects of *Gb* and *Ocs* extracts on brain water content in mice with PTZ-induced epilepsy. The assay demonstrated that PTZ significantly increased brain water content, and *Gb* and *Ocs* extracts administration reduced brain water content (Fig 3D). Although no significant differences were noted between the two concentrations of *Gb* and *Ocs* extracts, slightly lower water content was recorded at 16 g/kg than at 8 g/kg (Fig 3D).

## Effects of *Gb* and *Ocs* extracts on acetylcholine activity and neuronal loss-induced histological changes in the PTZ-induced epilepsy mouse model

The PFC and HC are crucial regions for social recognition, behavior, learning, and memory. In this study, we investigated the expression levels of acetylcholinesterase (AChE) and choline acetyltransferase (ChAT), the rate-limiting enzyme for acetylcholine synthesis and the primary metabolic enzyme of AChE, respectively. The western blot analysis of the PFC and HC regions of epileptic mice on D30 revealed higher AChE levels in the 40 mg/kg PTZ-treated group than in the control group (Fig 4A and 4B). The AChE levels were almost completely restored in the groups co-treated with 8 or 16 g/kg *Gb* and *Ocs* extracts (Fig 4A and 4B). In contrast, the ChAT levels in the PFC decreased in the 40 mg/kg PTZ-treated group and increased significantly in the groups treated with 8 or 16 g/kg *Gb* and *Ocs* extracts (Fig 4A). However, in the HC group, a low concentration (8 g/kg) of both insect extracts did not restore the PTZ-induced decrease in ChAT levels (Fig 4B). Additionally, a combined treatment with PTZ and VPA almost completely restored the expression levels of these two proteins to the range reported in the controls in both brain regions (Fig 4A and 4B).

The expression of the neuronal marker proteins, postsynaptic density protein-95 (PSD-95) and Neu-N, was determined to clarify the effect of *Gb* and *Ocs* extracts on PTZ-induced

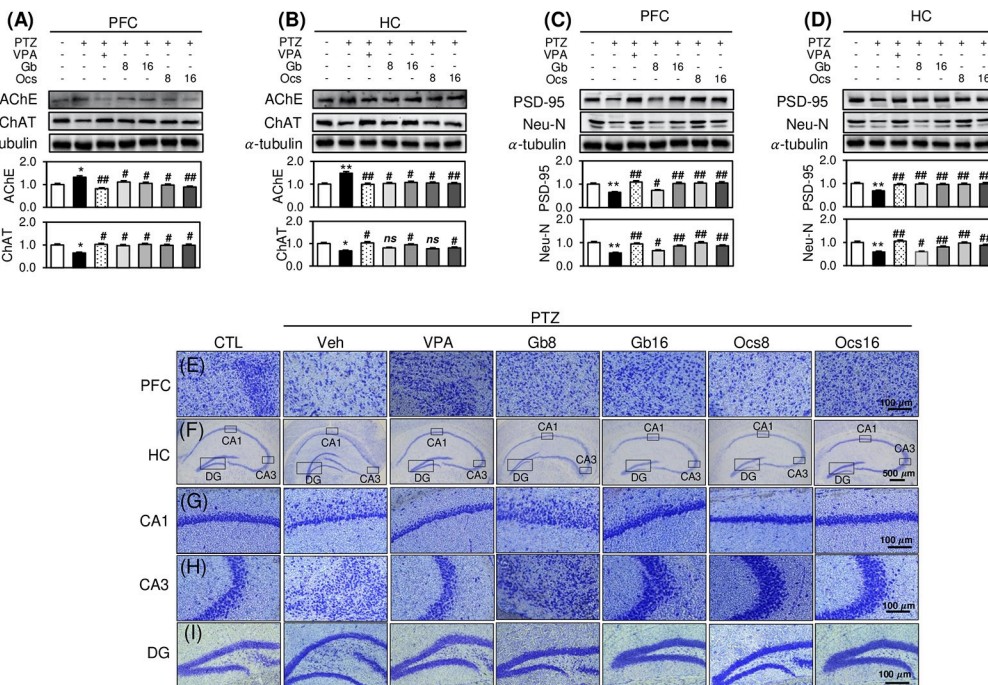

**Fig 4. Effects of *Gb* and *Ocs* extracts on enzymes associated with the regulation of acetylcholine activity and neuronal damage-induced histopathological changes in the mouse model of PTZ-induced epilepsy.** (A-D) Immunoblots for acetylcholinesterase (AchE), choline acetyltransferase (ChAT), postsynaptic density protein-95, and neuronal nuclear protein from the PFC and HC tissue lysates at D31. Bars denote a densitometric analysis of the immunoblots. (E-I) Representative sections of Nissl staining of different brain regions in each group at D31. Coronal brain sections (20 μm thick) passing through the prefrontal cortex (E) and HC (F), with the CA1 (G), CA3 (H), and dentate gyrus (I) stained with toluidine blue. Mean (n = 3) ± SD. *$P \leq 0.05$ and **$P \leq 0.01$ vs. control group; not significant, *ns*, #$P \leq 0.05$ and ##$P \leq 0.01$ vs. PTZ group.

neuronal viability. Western blotting demonstrated that PTZ significantly reduced the expression of these two proteins (Fig 4C and 4D). Combination treatments using *Gb* and *Ocs* extracts completely recovered the PTZ-mediated decrease in PSD-95 and Neu-N expression in both the PFC and HC (Fig 4C and 4D). However, low concentrations of *Gb* extract (8 g/kg) were ineffective (Fig 4A and 4B).

Similarly, to confirm whether *Gb* and *Ocs* extracts had histopathological effects on the PTZ-treated epilepsy mouse model, we performed Nissl staining. Brain tissue staining showed that PTZ treatment induced severe damage to the neural tissues in the PFC and HC (Fig 4E–4H). Among the imaged brain regions, the CA1 and CA3 HC showed the most severe neuronal loss (Fig 4G and 4H). In contrast, the DG of the HC showed no neuronal loss (Fig 4I). However, administering *Gb* and *Ocs* extracts almost completely recovered the PTZ-induced tissue damage, especially in the CA1 and CA3 regions of the HC (Fig 4G and 4H), but not in the DG (Fig 4I). In particular, the *Ocs* extract led to remarkable improvement than did the *Gb* extract, especially at 16 g/kg (Fig 4E–4H). In particular, the most significant protective effect after administering *Gb* and *Ocs* extracts was detected in the CA3 of the HC subregion (Fig 4H).

## Effects of *Gb* and *Ocs* extracts on altered activities of epileptogenesis-associated excitatory and inhibitory neurotransmitter receptors in PTZ-induced epilepsy mouse model

Epilepsy disturbs synaptic transmission by altering the balance between excitatory and inhibitory receptor activities. We further examined whether *Gb* and *Ocs* extracts reversed the PTZ-

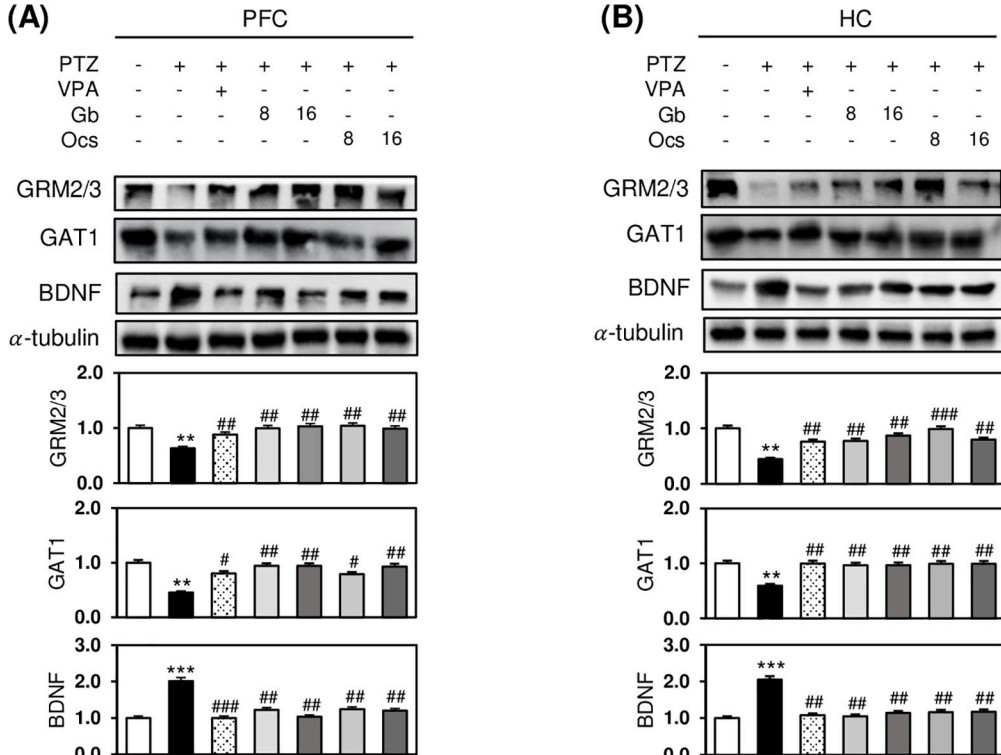

**Fig 5. Effects of *Gb* and *Ocs* extracts on excitatory and inhibitory neurotransmitter receptor-associated epileptogenesis in the PTZ-induced epilepsy mouse model.** (A, B) Immunoblots for GRM2/3, GAT1, and BDNF levels in the PFC (A) and HC (B) tissue lysates at D31 treated with different combinations of drugs. Bars indicate a densitometric analysis of the immunoblots. α-Tubulin was used as a loading control. Mean (n = 3) ± SD. **$P \leq 0.01$ and ***$P \leq 0.001$ vs. control group; *ns*, #$P \leq 0.05$, ##$P \leq 0.01$, and ###$P \leq 0.001$ vs. the PTZ group.

triggered abnormal activity of these two receptors. PTZ injection significantly increased BDNF expression but noticeably decreased GRM2/3 and GAT1 receptor expression in both the PFC and HC regions of the brain (Fig 5). Both doses of the insect extract (8 and 16 g/kg) showed similar levels of protein expression (Fig 5). However, combining insect extracts with PTZ almost completely reversed the effects of PTZ on the GRM2/3, GAT1, and BDNF receptors (Fig 5).

## Effects of *Gb* and *Ocs* on antioxidant defense system and apoptosis in the brains of the PTZ-induced epilepsy mouse model

Regarding the effects of *Gb* and *Ocs* extracts on PTZ-induced oxidative stress, compared with the control treatment, PTZ caused a significant decrease in GSH and SOD enzyme activity (Fig 6A and 6B), but increased MDA levels (Fig 6A–6C). Insect extracts along with PTZ significantly reversed the PTZ-induced disturbance of GSH and SOD activities as well as MDA levels (Fig 6A–6C), especially with higher restoration when using *Gb* treatment compared with when using *Ocs* (Fig 6A–6C). Moreover, oxidative stress was similarly mitigated by treatment with 8 and 16 g/kg insect extract (Fig 6A–6C).

Regarding the changes in apoptosis-associated signals examined using western blotting, PTZ treatment significantly increased the expression of Bax, c-PARP, and c-caspase 3 compared with that shown in the control group (Fig 7A and 7B). However, when we co-treated insect extracts from *Gb* and *Ocs* with PTZ, except for Bcl-XL expression in the group treated

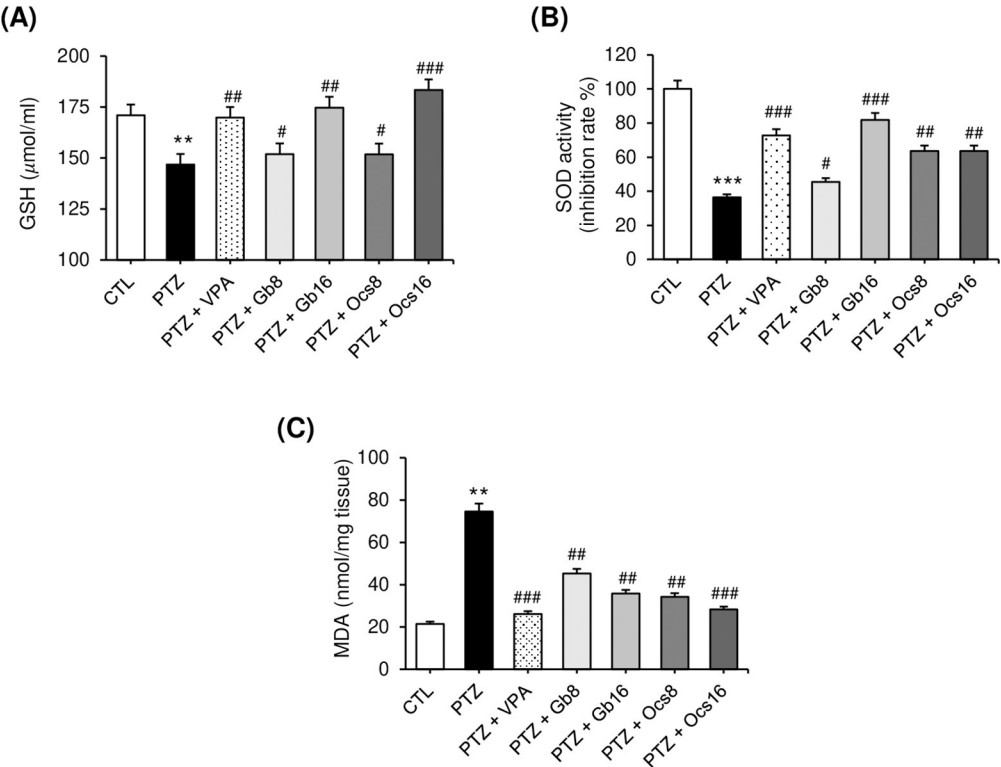

**Fig 6. Effects of *Gb* and *Ocs* extracts on antioxidant enzyme activity in the brain parenchyma of the PTZ-induced epilepsy mouse model.** (A-B) The quantitative measurement of antioxidant enzymes from whole-brain tissue lysates at D31 in each group treated with various combinations of drugs. (A) Reduced glutathione (GSH) and (B) superoxide dismutase (SOD) levels. Mean (n = 3) ± SD. **$P \leq 0.01$ and ***$P \leq 0.001$ vs. control group; #$P \leq 0.05$, ##$P \leq 0.01$, and ###$P \leq 0.001$ vs. the PTZ group.

with PTZ plus 8 g/kg *Gb* extract, the expression of Bcl-XL, Bax, c-PARP, and c-caspase 3 in the other groups was noticeably restored to the control levels (Fig 7A and 7B).

As shown in this study, PTZ activated ROS formation and lipid peroxidation. This contributes to the activation of MMP2 and the inhibition of tight junction proteins. Moreover, PTZ treatment induced neuronal hyperexcitability through altering the activity of excitatory and inhibitory receptors, such as GRM2/3, GAT1, and BDNF, as well as neuronal viability through Neu-N and PSD-95 suppression (Fig 8).

## Discussion

Epilepsy is a neurological disorder characterized by sudden onset of epileptic seizures [28]. This study evaluated the neuroprotective potential of *Gb* and *Ocs* extracts in mice with PTZ-induced epilepsy. Specifically, PTZ induced AChE upregulation in the PFC and HC of epileptic mice. AChE is involved in the metabolism of brain acetylcholine and is implicated in several cognitive disorders, including AD, schizophrenia, anxiety, narcolepsy, and epilepsy [29–31]. Acetylcholine is a neurotransmitter; therefore, drugs affecting cholinergic pathways can contribute to severe adverse effects, varying in intensity from convulsions to paralysis [32]. In the central nervous system, cholinergic projections from the basal forebrain to the cerebral cortex and HC contribute to cognitive function. In contrast, AChE in the peripheral nervous system promotes muscle activity and is a major neurotransmitter in the autonomic nervous system

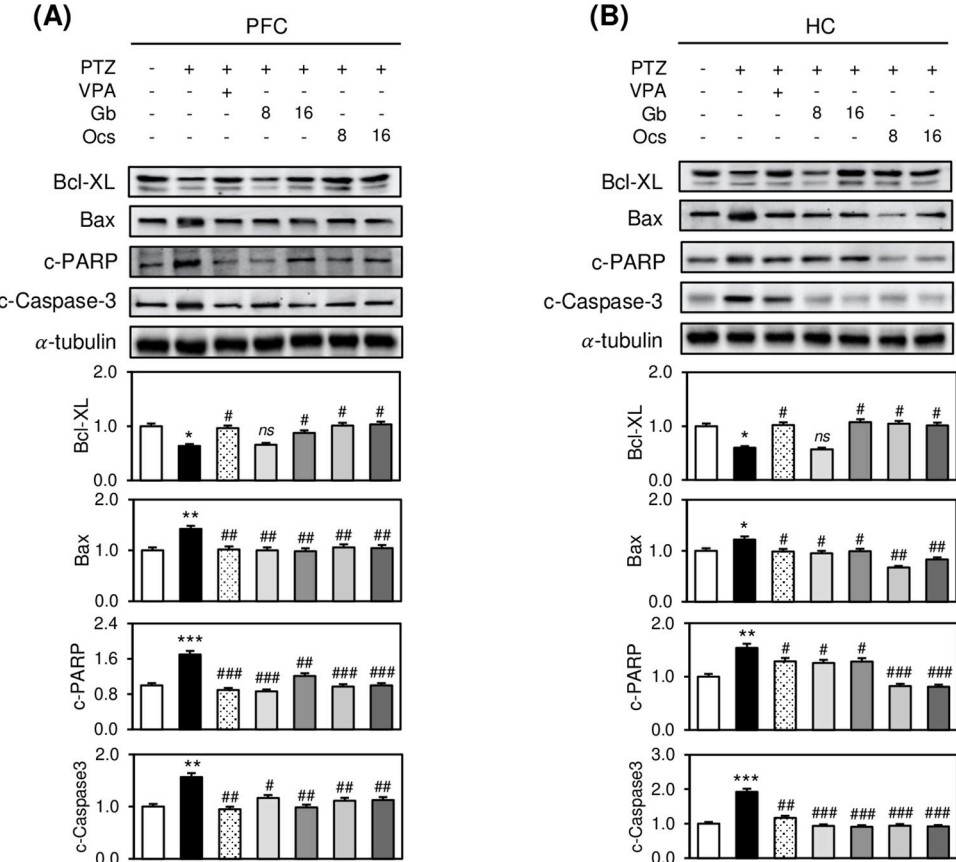

**Fig 7. Effects of *Gb* and *Ocs* extracts on apoptosis in the brain tissues of the PTZ-induced epilepsy mouse model.**
(A, B) Immunoblots of the PFC (A) and HC (B) tissue lysates at D31 treated with different combinations of drugs. Bars indicate the densitometric analysis of the immunoblots. α-Tubulin was used as a loading control. Mean (n = 3) ± SD. *$P \leq 0.05$, **$P \leq 0.01$, and ***$P \leq 0.001$ vs. control group; *ns*, #$P \leq 0.05$, ##$P \leq 0.01$, and ###$P \leq 0.001$ vs. PTZ group.

[33]. Previous studies have demonstrated that PTZ increases AChE activity and is associated with cholinergic neurotransmission [34].

This study demonstrated the effects of *Gb* and *Ocs* extracts on BBB disruption and brain edema, as well as on PTZ-induced neuronal damage and oxidative stress. BBB disruption severely affects hydraulic conduction functions, leading to the leakage of selective blood plasma components into the brain. BBB damage is considered one of the early causes that lead to a decline in the neurological and cognitive function of the brain, thereby disrupting its normal physiological function [35]. Moreover, BBB disruption causes cerebral edema and the pathogenesis of several neurological and neuropsychiatric disorders, such as AD, PD, stroke, epilepsy, and ASD [18,19,36].

Two insect extracts, *Gb* and *Ocs*, were shown to have a significant protective effect on several brain regions such as the HC and PFC, and this result was attributable to BBB damage, as shown by the reduced levels of Evans blue dye. Furthermore, the detrimental effects of PTZ were higher in the HC region than in the other brain regions. Several human and animal studies on epilepsy have reported that PTZ exposure causes BBB disruption and that increased BBB permeability is associated with the disruption of tight junctions or increased pinocytotic activity in barrier-type endothelial cells, subsequently leading to cerebral edema [37,38]. In this study, *Gb* and *Ocs* extracts significantly improved endothelial tight junction proteins such

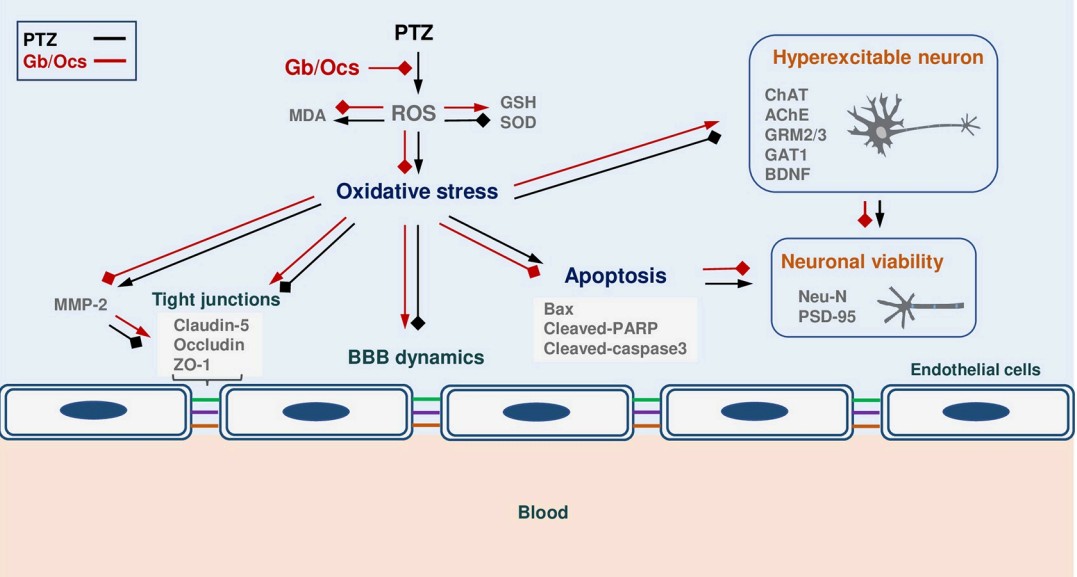

**Fig 8. Schematic illustration of the possible role of *Gb* and *Ocs* extracts in the PTZ-induced epilepsy mouse model.** PTZ stimulates ROS formation via the reduction of GSH and SOD, but increases lipid peroxidation. Increased ROS may promote oxidative stress, thus actively triggering multiple signaling pathways, including the activation of MMP2 and the inhibition of apoptosis and tight junction proteins, such as claudin-5, occludin, and ZO-1, partly contributing to BBB damage. Furthermore, PTZ-induced oxidative stress accelerates neuronal hyperexcitability via the abnormal regulation of ChAT, AChE, GRM2/3, GAT1, and BDNF. This leads to neuronal damage and cell death. Abbreviations: **PTZ**, pentylenetetrazole; **Gb**, *Gryllus bimaculatus*; **Ocs**, *Oxya chinensis sinuosa*; **ROS**, reactive oxygen species; **MDA**, malondialdehyde; **GSH**, glutathione; **SOD**, superoxide dismutase; **MMP-2**, matrix metalloproteinase-2; **BBB**, brain blood barrier; **ChAT**, choline acetyltransferase; **AChE**, acetylcholinesterase; **GRM2/3**, metabotropic glutamate receptor 2/3; **GAT1**, gamma-aminobutyric acid (GABA) transporter 1; **BDNF**, brain-derived neurotrophic factor; **Neu-N**, neuronal nuclear protein; **PSD-95**, postsynaptic density protein-95.

as claudin-5, ZO-1, and occludin, demonstrating the protective role of these bioactive materials in BBB integrity. Previous studies on animal models of epilepsy have indicated that MMPs are involved in epileptogenesis, apoptosis, and synaptic plasticity [39]. MMP-2 has been recognized for its potential role in the degradation of tight junction proteins and subsequent disruption of the BBB [40]. Therefore, in our study, the decreased expression of MMP-2 in epileptic mice after receiving *Gb* and *Ocs* extracts might have contributed to the restoration of the BBB permeability by controlling tight junction protein degradation in the PFC and HC.

Cognitive Decline in Epilepsy is associated with abnormal neuronal development. In this study, the PTZ-induced reduction in neuronal cells in the PFC and HC was restored to normal after administering *Gb* and *Ocs* extracts. Neurons play an essential role in signal propagation and neurodevelopment in the CNS. Oxidative stress is a critical factor in the pathophysiology of epilepsy and is associated with an intrinsic risk for neurodegeneration. Many studies have reported that PTZ increases oxidative stress in mice by decreasing the levels of GSH, CAT, and SOD and increasing the level of lipid peroxidation [41]. SOD and CAT are antioxidant enzymes that have important activities that alleviate the toxic effects of ROS [42]. Since neurons are more sensitive to ROS injury during the early stages of brain development, an imbalance between ROS production and scavenging systems leads to neuronal damage. ROS-induced cytotoxicity is closely associated with and mostly precedes apoptotic cell death [9,43]. Interestingly, *Gb* and *Ocs* extracts protected hippocampal neurons and suppressed apoptosis associated with seizure propagation. These extracts decreased the expression of apoptotic signaling molecules, such as Bax, caspase-3, and c-PARP, and increased the expression of anti-

apoptotic signaling molecules, such as Bcl-XL. Therefore, our findings suggest that PTZ induces normal neuronal cell viability via oxidative stress, ultimately resulting in apoptotic cell death via the breakdown of the ROS defense barrier.

Epileptogenesis occurs after a brain injury, during which hyperexcitable neural networks are formed. The typical pathophysiological characteristics of epileptogenesis include apoptosis, axonal sprouting, and alterations in neurotransmitter release and neurogenesis [44]. The expression of BDNF in both the PFC and HC of our epileptic mice may reflect one of these hyperexcitable neural responses. In particular, hyperexcitability likely occurs owing to the loss of inhibitory neurons, such as GABAergic interneurons, and the lack of the balancing ability of glutamatergic neurons with increased excitation, leading to reduced inhibition of GABA level [45–47]. In our study, the expression of mGluR2/3, a presynaptic glutamate receptor associated with the inhibition of presynaptic glutamate release, was significantly reduced after PTZ kindling in mice. Moreover, GAT1, a glutamate transporter confluent expressed in the nerve endings of GABAergic interneurons and astrocytes, was significantly reduced in both the PFC and HC regions of the brain in the PTZ-induced epileptic mice. Interestingly, this series of epileptogenesis observed in the brain of an epileptic mouse has been largely recovered with administering insect extracts, possibly reducing neuronal loss and seizure scores.

In this study, we reported that *Gb* and *Ocs* extracts attenuated brain edema and BBB damage by restoring tight junction proteins and MMP-2 in a PTZ-induced animal model of epilepsy. Although the precise target molecules and underlying mechanisms require further investigations, *Gb* and *Ocs* extracts protected neuronal cells by blocking a series of these reactions, leading to normal brain activity and development in mice with PTZ-induced epilepsy. Since the edible insect extracts used in this study were complex and not simple components, further studies are required to identify the efficacy of individual ingredients in the extracts, as well as the impact of these biochemical ingredients on pathological pathways. Similarly, since various cell types communicate in the brain, further in-depth research is necessary to determine which of the many types of brain cells, including astrocytes, microglia, oligodendrocytes, pericytes, and neurons, are affected by *Gb* or *Ocs* extracts as the sensitizer or initiator in the first defense system against PTZ-induced epilepsy in mice.

Nevertheless, in this study, we provided new insights into the prevention and treatment of epilepsy using natural bioactive substances that can regulate BBB homeostasis, an important physical barrier in the brain, and synaptic plasticity through the regulation of excitability and inhibition of nerve cells.

## Supporting information

**S1 File.**
(PDF)

**S1 Table. Percent mortality of mice used in PTZ-dose optimization.**
(DOCX)

**S2 Table. Percent mortality of mice used in the pilot study for optimization of insect extracts concentrations.**
(DOCX)

**S3 Table. Records of the onset of seizures with different concentrations of the insect extracts.**
(DOCX)

## Acknowledgments

We thank the Center for University-Wide Research Facilities at Jeonbuk National University for confocal microscopy and Editage (www.editage.co.kr) for the English language editing.

## Author Contributions

**Conceptualization:** Sook-Jeong Lee.

**Data curation:** Ngoc Buu Tran, Sook-Jeong Lee.

**Formal analysis:** Ngoc Buu Tran.

**Funding acquisition:** Sook-Jeong Lee.

**Investigation:** Sook-Jeong Lee.

**Methodology:** Ngoc Buu Tran.

**Project administration:** Sook-Jeong Lee.

**Software:** Ngoc Buu Tran.

**Supervision:** Sook-Jeong Lee.

**Validation:** Sook-Jeong Lee.

**Writing – original draft:** Ngoc Buu Tran, Sook-Jeong Lee.

**Writing – review & editing:** Sook-Jeong Lee.

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
