## [Decision Letter · Decision Letter 0]

4 Jun 2023

PONE-D-23-13972Effects of Gryllus bimaculatus and Oxya chinensis sinuosa extracts on brain damage through blood-brain barrier control and apoptosis in pentylenetetrazol-induced epilepsy in micePLOS ONE

Dear Dr. Lee,

Thank you for submitting your manuscript to PLOS ONE. After careful consideration, we feel that it has merit but does not fully meet PLOS ONE’s publication criteria as it currently stands. Therefore, we invite you to submit a revised version of the manuscript that addresses the points raised during the review process.

We look forward to receiving your revised manuscript.

Kind regards,

Ahmed E. Abdel Moneim

Academic Editor

PLOS ONE

Journal Requirements:

"Name of grant receiver: S-J Lee

Funder: National Research Foundation (NRF) grant funded by the Korean government

Grant Number: 2021R1A2C1005980"

"This work was supported by a National Research Foundation (NRF) grant funded by the Korean government (MSIT) (Grant No. 2021R1A2C1005980). This study was also supported by Jeonbuk National University, Republic of Korea. Mr. Ngoc Buu Tran was supported by the Brain Korea 21 program at the Department of Bioactive Material Sciences. We would like to thank eWorldediting (www.eworldediting.com) for English language editing."

"Name of grant receiver: S-J Lee

Funder: National Research Foundation (NRF) grant funded by the Korean government

Grant Number: 2021R1A2C1005980"

"I have read the journal's policy and the authors declare no conflicts of interest."

Reviewers' comments:

Reviewer's Responses to Questions

**Comments to the Author**

1. Is the manuscript technically sound, and do the data support the conclusions?

Reviewer #1: Yes

Reviewer #2: Yes

2. Has the statistical analysis been performed appropriately and rigorously? 

Reviewer #1: Yes

Reviewer #2: Yes

3. Have the authors made all data underlying the findings in their manuscript fully available?

Reviewer #1: Yes

Reviewer #2: Yes

4. Is the manuscript presented in an intelligible fashion and written in standard English?

Reviewer #1: Yes

Reviewer #2: Yes

5. Review Comments to the Author

Reviewer #1: A mouse model of PTZ-induced epilepsy was used in this study to examine the effects of extracts from the edible insects Gryllus bimaculatus (Gb) and Oxya chinensis sinuosa (Ocs). The work focused on how Gb and Ocs extracts affected BBB permeabilization and neuronal death brought on by oxidative stress.

This is interesting study.

Remarks:

According to the data it seems that the protective role of insect extract is attributed to antioxidant and antiapoptotic effects characteristic of the extract. However, no oxidative stress markers were estimated (MDA or 4-HNE) or carbonyl proteins or ROS. Please clarify.

Reviewer #2: Evaluation of Effects of Gryllus bimaculatus and Oxya chinensis sinuosa extracts on brain damage through blood-brain barrier control and apoptosis in pentylenetetrazol-induced epilepsy in mice

Authors have evaluated the potential neuroprotective role of Gb and Ocs extracts on epileptic seizures model induced by PTZ in mice through evaluating the BBB integrity and oxidative damage-induced neuronal apoptosis.

The topic is, in my opinion, an interesting one, and the manuscript is well organized and written smoothly, and easy to follow. However, for a better report, the authors need to consider the following minor points:

• Please add the examined doses and duration of treatments to the abstract.

• Line 64: please use only ROS, as you aleady identified the full name in line 54.

• Please address the concerns about the novelty of your manuscript very thoroughly.

• Please explain how authors determined the numbers of mice in this study. Please justify the numbers.

• Do the authors record any mortality among the animals in this study?

• On what basis authors have selected the doses of the extracts (8 and 16 g/kg)?

• Please mention the examined brain regions under construction of the PTZ-induced mouse model of epilepsy and epileptogenesis assessment

• Have the authors tested their data normality?

• Why authors have used Fisher’s least instead of Duncan or Tukey tests?

• The relationships among the indicators detected in this study should be more addressed.

• I would suggest authors to evaluate the major excitatory and inhibitory amino acids in the examined brain regions as they play essential role in epileptogenesis.

• Please add limitations for this work

6. PLOS authors have the option to publish the peer review history of their article (what does this mean?). If published, this will include your full peer review and any attached files.

Reviewer #1: **Yes: **M.A. El-Missiry

Reviewer #2: **Yes: **Rami B. Kassab

---

## [Author Response · Author response to Decision Letter 0]

13 Jul 2023

Response to Reviewers

Reviewer #1: 

We appreciate the constructive criticism and helpful comments from the reviewers. We have done our best to address the issues raised by the reviewers. 

The following aspects noted by the reviewer were addressed:

Specific Comments:

1. According to the data, it seems that the protective role of insect extract is attributed to antioxidant and antiapoptotic effects characteristics of the extract. However, no oxidative stress markers were estimated (MDA or 4-HNE) or carbonyl proteins or ROS. Please clarify.

Response: Thank you for your comment. Based on the suggestion, we performed a lipid peroxidation (MDA) assay to further confirm the relevance of oxidative stress in edible insect-fed mice with epilepsy (Figure 6C).

Reviewer #2: 

We appreciate the reviewer’s detailed criticism and helpful comments. We have attempted to address the reviewers’ comments in the revised manuscript. 

Following are responses to specific points.

1. Please add the examined doses and duration of treatments to the abstract.

Response: We sincerely apologize for the lack of detailed descriptions of the experimental methods. As per your suggestions, we have rewritten the abstract with a description of the doses and duration of insect extract treatment (lines 24–25). 

2. Line 64: Please use only ROS, as you already identified the full name in line 54.

Response: As the reviewer pointed out, we corrected “reactive oxygen species” to “ROS” (line 66). 

3. Please address the concerns about the novelty of your manuscript very thoroughly.

Response: Thank you for your comment. Epilepsy can be detrimental to the brain. In this study, we focused on the effects of new bioactive materials, which are edible insects, on epilepsy-induced blood-brain barrier (BBB) and neuronal damage in mice. Edible insects, the materials used in our study, are gaining attention as a future food source owing to the associated low-carbon emissions considering the global climate crisis and are especially in the spotlight as nutritional foods comprising higher protein, lower fat, and higher mineral content than in livestock and poultry. Furthermore, research on the efficacy of edible insects against various diseases has drawn considerable attention. This study confirmed the effects of edible insect extracts on BBB damage and nerve cell damage associated with oxidative stress in epilepsy. Our results show the possibility that extracts derived from eco-friendly natural materials can be developed as important biological materials for improving and treating epilepsy.

We have addressed the novelty of our study in the Introduction and Discussion sections (lines 73-78, lines 471–480).

4. Please explain how authors determined the numbers of mice in this sutdy. Please justify the numbers.

Response: We used 8–10 mice per group for the experiments. Eight mice were monitored in order to determine the optimal dose of PTZ, and behavioral scoring was statistically analyzed. Ten mice per group were used to examine the combined treatment using the insect extracts. The number of mice used for each experiment was added to the Methods section and was organized in detail in Supplementary Tables 1, 2, and 3 (S1–S3 Tables). 

5. Do the authors record any mortality among the animals in this study?

Response: Yes, we recorded the detailed mortality of the mice after each experiment. We have added the summarized mortality of mice from each experiment to Supplementary Tables 1, 2, and 3 (S1–S3 Tables).

6. On what bais authors have selected the doses of the extracts (8 and 16 g/kg)?

Response: Thank you for your comment. Prior to inducing epilepsy to generate the mouse model, many studies using insect extracts on mice and rats have been revised. After selecting several concentrations of insect extracts (8, 16, and 20 g/kg), seizure onset time, seizure duration, and mortality were analyzed. Although 20 g/kg of insect extract effectively protected mice with PTZ-induced epilepsy from seizure onset time and seizure duration, mortality was higher than in mice treated with PTZ alone. Therefore, we chose 8 and 16 g/kg insect extracts for the experiments. The results of these preliminary experiments are summarized in detail in Supplementary Table 3 and described in the Results section (lines 217–221).

7. Please mention the examined brain regions under construction of the PTZ-induced mouse model of epilepsy and epileptogenesis assessment.

Response: As the reviewer suggested, we have mentioned the brain regions used for epileptogenesis assessment in the Methods section (lines 144 to 146, 151, 157, 170, 184-186).

8. Have the authors tested their data normality?

Response: Thank you for your comment. After conducting the animal experiment, we thoroughly evaluated each experimental result, carefully analyzed, and statistically evaluated the figures. The sample data were representative, within acceptable limits, and followed a normal distribution pattern.

9. Why authors have used Fisher’s least instead of Duncan or Tukey tests?

Response: Thank you for your valuable comment. All three mentioned are post hoc tests used to determine which differences between pairs of group means are significant. The Fisher’s least significant difference test, also called the protected t test, is the most liberal post hoc test, as it has the lowest critical value, which increases the power to detect an effect or mean difference between the two groups. This did not require the number of participants in each group to be equal. The Tukey’s Honestly Significant Difference test is a more conservative test, where the critical levels are larger for each pairwise comparison. Both tests produced the same results and were used when deciding whether to retain or reject the null hypothesis. The Waller–Duncan k-ration t test is a multiple range test. Unlike the Tukey’s test, this test is not based on the principle of controlling for Type I errors; instead, it compares Type I and II error rates based on Bayesian principles. Tukey’s HSD is the most conservative, and Fisher’s LSD is the most liberal, as shown in published research studies. In this study, we aimed to avoid numerical heterogeneity between groups by assessing the percentage mortality in vivo. In addition, the Fisher’s test is a common and widely used test with good results and a few disadvantages affecting the evaluation of the study. Therefore, the Fisher’s test was used to analyze our experimental results and ongoing studies statistically.

10. The relationships among the indicators detected in this study should be more addressed.

Response: Based on the reviewer’s suggestion, we have addressed the relationships between the indicators detected in this study in the Results section (lines 378-382). Moreover, we have created a new figure (Figure 8) depicting the possible relationships between signaling molecules associated with Gb/Ocs extract-mediated protective effects in PTZ-triggered epileptic mice. 

11. I would suggest authors to evaluate the major excitatory and inhibitory amino acids in the examined brain regions as they play essential role in epileptogenesis.

Response: We agree with the reviewer’s comment. In line with the suggestion, we evaluated the representative receptor protein expression of each excitatory and inhibitory transmitter, such as metabotropic GluR2/3 receptor, gamma-aminobutyric acid transporter 1, and brain-derived neurotrophic factor receptor, in different brain regions by western blotting and confirmed how these excitatory and inhibitory receptors were expressed in our model system and how insect extracts influenced epileptogenesis-induced excitatory and inhibitory receptor expressions as well. These results have been added to Figure 5.

12. Please add limitations for this work.

Response: According to the reviewer’s comment, we have added the limitations of this study to the Discussion section (lines 471-480).

---

## [Decision Letter · Decision Letter 1]

23 Aug 2023

Effects of Gryllus bimaculatus and Oxya chinensis sinuosa extracts on brain damage via blood-brain barrier control and apoptosis in mice with pentylenetetrazol-induced epilepsy.

PONE-D-23-13972R1

Dear Dr. Lee,

We’re pleased to inform you that your manuscript has been judged scientifically suitable for publication and will be formally accepted for publication once it meets all outstanding technical requirements.

Kind regards,

Ahmed E. Abdel Moneim

Academic Editor

PLOS ONE

Additional Editor Comments (optional):

Reviewers' comments:

Reviewer's Responses to Questions

**Comments to the Author**

1. If the authors have adequately addressed your comments raised in a previous round of review and you feel that this manuscript is now acceptable for publication, you may indicate that here to bypass the “Comments to the Author” section, enter your conflict of interest statement in the “Confidential to Editor” section, and submit your "Accept" recommendation.

Reviewer #1: (No Response)

2. Is the manuscript technically sound, and do the data support the conclusions?

Reviewer #1: Yes

3. Has the statistical analysis been performed appropriately and rigorously? 

Reviewer #1: Yes

4. Have the authors made all data underlying the findings in their manuscript fully available?

Reviewer #1: Yes

5. Is the manuscript presented in an intelligible fashion and written in standard English?

Reviewer #1: Yes

6. Review Comments to the Author

Reviewer #1: (No Response)

7. PLOS authors have the option to publish the peer review history of their article (what does this mean?). If published, this will include your full peer review and any attached files.

Reviewer #1: **Yes: **Mohamed Amr ElMissiry

---

## [Editor Report · Acceptance letter]

1 Sep 2023

PONE-D-23-13972R1 

Effects of *Gryllus bimaculatus* and *Oxya chinensis sinuosa* extracts on brain damage via blood-brain barrier control and apoptosis in mice with pentylenetetrazol-induced epilepsy 

Dear Dr. Lee:

I'm pleased to inform you that your manuscript has been deemed suitable for publication in PLOS ONE. Congratulations! Your manuscript is now with our production department. 

Kind regards, 

on behalf of

Dr. Ahmed E. Abdel Moneim 

Academic Editor

PLOS ONE